# OpenReview forum: "A COMPREHENSION ON ETHICAL HACKING:  TOOLS, BOOKS, AND WEBSITES"
_IEEE.org/ICIST/2024/Conference — IEEE ICIST 2024 Conference Submission_

### Official Review · Reviewer_MzJw · 2024-08-29
**comment**

**Rating:** 4
**Confidence:** 4

**Review:**

This document lists tools, books, and websites that are extremely helpful for the ones looking to probe into ethical hacking. However, in my own opinion, the literature listed for ethical hacking is not sufficiently enough. Moreover, the architectures given only include parts of the tools. Authors should make a more detailed literature review on this topic, and more detailed future topics will be better for readers.

---

### Official Review · Reviewer_KRGF · 2024-09-02
**This paper should be modified carefully.**

**Rating:** 6
**Confidence:** 4

**Review:**

This paper lists tools, books, and websites that are extremely helpful for the ones looking to probe into ethical hacking.
The reviewer's comments are as follows:
1.The English grammar and format of this manuscript could be further polished and checked carefully.
2.The format of references is not uniform.
3.To enhance the quality of the manuscript, it is recommened refining the language to improve readability and ensuring that the concepts are communicated accurately.

---

### Official Review · Reviewer_jWwG · 2024-09-03
**This paper can be considered for publication.**

**Rating:** 6
**Confidence:** 2

**Review:**

The authors in this paper introduces their comprehension on ethical hacking tools, books, and websites. The reviewer has the following comments.
1. The presentation quality of this manuscript should be greatly improved.
2. The conclusion is too long, which makes it unreadable.

---

### Decision · Program_Chairs · 2024-09-06

Accept (Oral)